

# Technical Note: Advances in flash flood monitoring using UAVs

MT Perks[1]*, AJ Russell[1], ARG Large[1]

[1]Newcastle University, School of Geography, Politics and Sociology, Daysh Building, Claremont Road, Newcastle-upon-Tyne, NE1 7RU

*Correspondence to*: MT Perks (matthew.perks@ncl.ac.uk)

**Abstract** UAVs have the potential to capture information about the earth's surface in dangerous and previously inaccessible locations. Through image acquisition of flash flood events and subsequent object-based analysis, highly dynamic and oft-immeasurable hydraulic phenomenon may be quantified at previously unattainable spatial and temporal resolutions. The potential for this approach to provide valuable information about the hydraulic conditions present during dynamic, high-energy
flash floods has until now not been explored. In this paper we adopt a novel approach, utilising the Kande-Lucas-Tomasi (KLT) algorithm to track features present on the water surface which are related to the free-surface velocity. Following the successful tracking of features, a method analogous to the vector correction method has enabled accurate geometric rectification of velocity vectors. Uncertainties associated with the rectification process induced by unsteady camera movements are subsequently explored. Geo-registration errors are relatively stable and occur as a result of persistent residual distortion
effects following image correction. The apparent ground movement of immobile control points between measurement intervals ranges from 0.05 - 0.13m. The application of this approach to assess the hydraulic conditions present in Alyth Burn, Scotland during a 1:200 year flash flood resulted in the generation of an average 4.2 measurements m$^{-2}$ at a rate of 508 measurements s$^{-1}$. Analysis of these vectors provide a rare insight into the complexity of channel-overbank interactions during flash floods. The uncertainty attached to the calculated velocities is relatively low with a spatial average across the area of ± 0.15m s$^{-1}$.
Little difference is observed in the uncertainty attached to out-of-bank velocities (± 0.15m s$^{-1}$), and within-channel velocities (± 0.16m s$^{-1}$), illustrating the consistency of the approach.

## 1 Introduction

The occurrence of flash flooding from intense rainfall in Western Europe is predicted to increase throughout first half of the 21st Century (Beniston, 2009;Rojas et al., 2012). These events pose severe risks to society, transform communities and under
extreme conditions can permanently alter the state of the river system (Milner et al., 2013;Doocy et al., 2013). Flash floods in fluvial systems pose high risks to communities especially when they occur in small, upland catchments where orographic effects can enhance precipitation intensity with runoff being concentrated rapidly along narrow and steep flow pathways (Garambois et al., 2014;Sangati et al., 2009;Bracken and Croke, 2007). Despite a substantial body of work on physical flood processes observed in research catchments (e.g. Quinn and Beven, 1993;Mayes et al., 2006;Soulsby et al., 2000), there is
currently a paucity of data describing the antecedent and concurrent processes associated with extreme flash flood events. This is mainly due to conventional monitoring networks often failing to adequately sample small, responsive catchments (Gaume



and Borga, 2008;Soulsby et al., 2008;Braud et al., 2014;Borga et al., 2008). Measurement and monitoring of these events is therefore largely responsive rather than active, opportunistic rather than strategic, and hindered by practical difficulties (Borga et al., 2008;Tauro et al., 2015b). Making observations of peak flood discharge ($Q_{peak}$) in real-time remains a significant practical challenge.

Given current operational constraints, favourable sources of process-data during flash floods and particularly at $Q_{peak}$ in ungauged catchments often rely on *post-hoc* analyses of air and space borne earth observation sensors (e.g. visible, near-infrared and multispectral imaging and synthetic aperture radar). Increasing availability of these remote sensed data has furthered our understanding of floodplain inundation processes (e.g. Wright et al., 2008); enabled hydraulic properties such as

roughness (Simeonov et al., 2013), river stage and discharge (Liu et al., 2015) to be successfully modelled; provided justification for the incorporation of spatially and temporally varied roughness values (Schumann et al., 2007;Mason et al., 2003); and enabled calibration and validation of hydrodynamic models (e.g. Martinis et al., 2009;Refice et al., 2014). Various contributions have been enabled by the fortuitous availability of archived satellite and aerial records (e.g. Kääb and Leprince, 2014;Chen and Mied, 2013). However, the highly transient temporal and spatial domains of flash floods, combined with the

significant lead times required to mobilise monitoring resources, has up until now limited the use of archived satellite and aerial records to larger, more slowly responding catchments (e.g. Wong et al., 2015).

The widespread availability of unmanned aerial vehicles (UAVs) has, in recent years, increased our ability to monitor and quantify higher magnitude, lower frequency environmental phenomena (e.g. Niethammer et al., 2012;Ryan et al., 2015), whilst

at the same time reducing operational costs of traditional environmental monitoring (Fekete et al., 2015). The potential for the use of UAVs for non-contact flow measurement has been identified (Kääb and Leprince, 2014), leading to proof-of-concept studies utilising UAVs for monitoring of low-flow conditions (e.g. Patalano et al., 2015;Tauro et al., 2015a;Pagano et al., 2014;Detert and Weitbrecht, 2015;Tauro et al., 2015b). However, the potential for this approach to provide valuable information about the hydraulic conditions present during dynamic, high-energy flash floods has yet to be realised.

Image based non-contact methods of flow estimation utilise algorithms designed to track optically visible features of the free-surface to determine the rate of fluid flow in artificial, or natural open-channels (Pentari et al., 2014;Jodeau et al., 2008;Kim et al., 2008;Sun et al., 2010;Le Boursicaud et al., 2015;Le Coz et al., 2010;Dramais et al., 2011;Puleo et al., 2012). The rate at which naturally occurring features (e.g. foam, seeds, woody debris and turbulent structures) or artificially introduced tracers

(e.g. Ecofoam chips, fluorescent micro-spheres, etc.) are displaced downstream can be used to estimate the free-surface velocity, which may be related to depth-averaged flow characteristics (e.g. Simeonov et al., 2013;Le Boursicaud et al., 2015;Fujita and Kunita, 2011;Dramais et al., 2011;Jodeau et al., 2008). Conceptually, terrestrial and airborne tracking of surface water features are similar; however the uncertainties associated with rectification of captured images to account for perspective, radial, and tangential distortions are compounded when using a UAV for image acquisition.  This is due to



unsteady camera movement, which must be accounted for if accurate geometric rectification of velocity vectors or oblique images is to be achieved (Kantoush et al., 2008;Kim et al., 2008). A second source of uncertainty is introduced in situations where low seeding densities prevail resulting in a lack of stable and identifiable surface features (Lewis and Rhoads, 2015). However in the case of flash floods, coherent flow structures at the free-surface and presence of washed-in floating material
may produce favourable seeding conditions (Dramais et al., 2011;Jodeau et al., 2008).

This paper presents a novel methodology for the derivation of key hydraulic data during flash floods using imagery captured by a low-cost, commercially available UAV platform. Our approach overcomes uncertainties associated with image rectification, transformation and feature tracking to determine river surface velocity during flash floods.  Our approach yields
fundamental process data, invaluable for flash flood reconstruction in ungauged river catchments. The adoption of this technique has the potential to significantly advance our understanding of high flow stage processes during flash floods.

## 2 Materials and Methods

### 2.1 Primary data collection

A Phantom Vision 2 UAV, equipped with a FC200 camera unit was deployed over Alyth Burn, Perthshire, Scotland (324600,
748600 OS BNG) on 17th July 2015 at ~11:00 BST. At this time, the river breached its banks as a result of a prolonged period of rainfall over the catchment. While rainfall totals were not in themselves extreme (41mm over a 6 hour period), the prolonged nature of the precipitation event coupled with the particular catchment configuration upstream of the town, resulted in over 70 properties being flooded and four footbridges in the town centre being destroyed (Perth & Kinross Council et al., 2015). Footage of the event was collected at 960 x 540 pixel (px) resolution at an acquisition rate of 25 frames per second (FPS).
Ground control points (GCPs) for the area of interest were required to convert the image (px) co-ordinates into geographical co-ordinates (OS BNG m). The deployment of a Leica MS50 multi-station shortly after the flood event enabled the generation of a detailed point cloud with an average point spacing of <0.002m from which GCPs could be accurately identified. These GCPs represented immobile objects that were present during the recording, and which persisted following the clean-up operation (e.g. lamp-posts and wall corners). Individual point clouds were joined using *CloudCompare* v2.6.1 (2015), resulting
in an internal error (RMS) of 0.04m. This point cloud was rectified to real-world co-ordinates through comparison with control point measurements (*n* = 12) obtained by a Leica GS14 GNSS system. This resulted in an additional error of 0.06m.

### 2.2 Camera Motion and Calibration

Due to the lack of available navigation data for the UAV, its starting position was modelled using an *a-priori* assumption about
its approximate location [$X_{est}$, $Y_{est}$, $Z_{est}$]. This was based on a visual assessment of the objects within view of the camera. 20,000 co-ordinate solutions were randomly generated ($X_{est} \pm 7.5$m; $Y_{est} \pm 7.5$m; $Z_{est} \pm 5$m) resulting in 8.9 discrete locations





per m$^3$. For each of these potential starting positions, a distorted camera model was generated (cf. Messerli and Grinsted, 2015). For each camera model, the radial distortion coefficients and image centre parameters that define the camera lens were fixed based on manufacturer's specification (DJI, 2015). The focal length, and view direction (yaw, pitch and roll) were however free parameters and allowed to vary accordingly. These were optimised to minimise the square projection error of the GCPs using a modified Levenberg–Marquardt algorithm (Fletcher, 1971). The optimal solution was subsequently defined as the master camera model, which was used as the basis for future projective transformations.

Figure 1. An example of georectification and projection of GCP positions (red dots) following optimisation of the distorted camera model alongside the location of actual GCP positions (blue dots).

Following optimisation of the UAVs starting location, prominent features and GCPs were tracked iteratively between subsequent frames using the Kande-Lucas-Tomasi (KLT) algorithm (Shi and Tomasi, 1994). Every $n$th and $n + 9$th frame, where $n$ equals the start of the tracking sequence, the start and finish positions of the successfully tracked features were stored in pixel units creating virtual velocity vectors representing motion during the previous 0.4s of video. These virtual velocity vectors were then corrected for background image displacement so that stationary objects yield zero or negligible velocity values. This was achieved using an approach analogous to the Vector Correction Method (Fujita and Kunita, 2011). This required the generation of an optimised camera model solution based on updated GCP co-ordinates for each frame (Messerli and Grinsted, 2015). This was achieved by randomly generating 1000 new positions proximal to the co-ordinates of the optimised model for the previous frame (X ± 0.25m;   Y ± 0.25m; Z ± 0.25m). These co-ordinates were then fixed whilst view direction was perturbed. The optimum model was produced by minimising the difference between the actual and projected GCP co-ordinates. Once camera movement is accounted for, the corrected virtual velocity vectors are converted to real-world start and finish co-ordinates giving $[X_n, Y_n]$ and $[X_{n+9}, Y_{n+9}]$ respectively. This was achieved through projection of the 2-D image pixel co-ordinates using a two-dimensional transformation (Fujita and Kunita, 2011; Fujita et al., 1998), based on the optimised camera models specific to $n$th and $n + 9$th frame (Messerli and Grinsted, 2015). During this process, features were only tracked if they lie within the central 90% of the image. This was necessary to minimise the potential for residual distortion effects to bias measurements, as these were most likely to persist close to the image boundaries (Detert and Weitbrecht, 2015). This process enables the calculation of 2-D velocities $[u, v]$ following application of a conversion factor $k$ to account for the number of tracked frames $I$ and seconds per frame $F$:

$$[\Delta X, \Delta Y] = [X_{n+9}, Y_{n+9,}] - [X_n, Y_n]$$  (1)

$$k = \frac{1}{(F \ I)}$$  (2)

$$[u, v] = [\Delta X, \Delta Y][k]$$  (3)





The degree to which the geo-rectification process is a success is assessed by comparing how the co-ordinates of the surveyed GCPs $[N, E]$ compare to the projected GCP locations $[N_T, E_T]$. The residuals $[r, s]$ represent the absolute positional error of the GCPs and provide a direct measure of the accuracy of the geometric transformation from pixel units into geographical co-ordinates, given by the Euclidean distance between the actual and projected locations $R_{EN}$ (Detert and Weitbrecht, 2015):

$$[r, s] = [N_T, E_T] - [N, E] \tag{4}$$
$$R_{EN} = (r^2 + s^2)^{0.5} \tag{5}$$

The degree to which the projection of the GCPs varies over time is assessed by examining the relative changes in the GCP projection locations (m) between the beginning and end of the feature tracking process:

$$[u_{EN}, v_{EN}] = [r_{n+9} - r_n), (s_{n+9} - s_n)] \tag{6}$$
$$U_{EN} = (u_{EN}^2 + v_{EN}^2)^{0.5} \tag{7}$$

2-D natural neighbour interpolation of the GCP errors is performed, giving spatially distributed estimates of $R_{EN}$ and $U_{EN}$. Velocity vectors in areas defined as having poor transformation accuracy (i.e. $\geq 1$m), or significant apparent movement of the GCPs between frames (i.e. $\geq 0.3$m) are removed prior to analysis, in addition to tracked features exhibiting minimal displacement (i.e. $\leq 0.3$m). This resulted in 48% of the original velocity vectors being eliminated. Data was not subject to any additional filtering.

## 3. Results

### 3.1 Camera Motion

Using the 20,000 potential solutions, the optimised master camera model was selected based on the minimum square projection error of the GCPs (RMSE). In this instance, the minimum RMSE was 11.4px ($n = 8$). Following the perturbation of geographical and orientation parameters for each frame, the flight path of the UAV was successfully modelled. Cumulative Euclidean distance travelled over the 140 frames was 13.2m (2.5m s$^{-1}$) whilst the camera rotated on the y-axis by 28°. During the video the RMSE of the optimised camera did not exceed 12.9px with a mean μ of 9.6px and a standard deviation σ of 1.3px.

Table 1. Optimised parameters of the distorted camera models





### 3.2 Positional Accuracy

Analysis indicates that the precision of the geometric projection $R_{EN}$ remains relatively stable throughout the video (Figure 2a). However the number of GCPs does exert some influence on the associated $R_{EN}$ value. The minimum $R_{EN}$ value of 0.4m is observed at 0.8s when 6 GCPs are within shot. With the removal of GCPs that are difficult to resolve, located close to the

upper edge of the frame, $R_{EN}$ naturally decreases. The maximum $R_{EN}$ value is 0.76m which occurs at 1.6s (13 GCPs). This provides an indication of the minimum spatial scale over which measurements should be averaged and reported. Significant spatial variability in $R_{EN}$ values are observed with median individual GCP $R_{EN}$ values ranging from $0.27 - 1.0$m (Figure 2b). However, the interquartile range of $R_{EN}$ for each GCP is relatively small, with a median value of 0.15m Furthermore, due to the lack of correlation between geolocation errors and the distance of the GCP from the camera source, we eliminate the

potential for significant errors being a function of reduced pixel density per unit area as GCP distance increases (Figure 2b). These findings indicate that the geo-registration errors are relatively stable and occur as a result of persistent residual distortion effects following image correction, especially close to the image boundaries, due to the specified transformation parameters being sub-optimal.

Figure 2. Box plots showing how projection residuals $R_{EN}$ (m) of all GCPs vary with: a) time; and b) distance from the UAV camera. Dot within circle = median; box = 25th and 75th percentiles; whiskers = extremes, open circle = outliers. Line = Number of GCPs/Distance of GCP from image source (m).

Whilst accurate geometric projection is essential for observed velocities to be assigned an appropriate spatial reference, the

precision of the transformation over time is of greatest importance (cf. Eq. 7). Unacceptable apparent ground velocities as a result of unstable transformation over time would undermine the value of tracking surface features. This error $U_{EN}$ is quantified by computing the relative movement of reference features across each tracking interval. Unaccounted for movement generally decreases over time following the maximum $U_{EN}$ of 0.28m at 1.2s through to the minimum of 0.05m at 2.4s (Figure 3a). Median $U_{EN}$ values continue to be $< 0.15$m throughout the sequence until the final frame when median $U_{EN}$ increases to 0.26m.

Unlike the spatial variability of $R_{EN}$ values, $U_{EN}$ values for specific GCPs are observed to be relatively consistent (Figure 3b). The median of the 15 GCPs ranges from 0.05 - 0.13m with no apparent relationship between the distance of the GCP and $U_{EN}$. These findings illustrate the relative spatial and temporal stability of the geometric transformation. Occasionally however the apparent velocity of fixed targets, and therefore associated error, is significant (i.e. $> 0.3$m). In these instances, features tracked within areas of unaccounted for movement are identified and filtered from subsequent analysis.

Figure 3. Box plots showing how the apparent movement $U_{EN}$ (m) of all GCPs varies with: a) time; and b) distance from the UAV camera. Dot within circle = median; box = 25th and 75th percentiles; whiskers = extremes, open circle = outliers. Line = Number of GCPs/Distance of GCP from image source (m).





### 3.3 Feature tracking & velocity estimation

Following the analysis of the 5.2s video, and the filtering of features tracked from within inaccurately projected regions of the image, a total of 2644 velocity vectors were compiled within a 624m$^2$ area of Alyth Burn and the surrounding inundated landscape. This results in an average of 4.2 measurements m$^{-2}$ at a rate of 508 measurements s$^{-1}$. Analysis of these vectors

provides an insight into the complexity of interactions between flow, sediment load and debris during flash floods. The bridge in the video (which was ultimately destroyed) was recorded in the imagery as being blocked by coarse woody debris. Due to the turbulent vortices generated by this blockage, surface velocities upstream of the bridge are calculated to be minimal (0.3 – 0.4 m s$^{-1}$). This blockage reduced conveyance of the flood waters with a proportion of channel flow becoming diverted into the adjacent street where surface velocities exceeded 1.2m s$^{-1}$. Similar breaches of the river's defences upstream of the camera

frame result in the routing of flood waters along the adjacent street. This routing produces velocities in the region of 0.9m s$^{-1}$ before these waters are mixed with those diverted from the main channel at the bridge within the camera shot. Further along the road, flow is disrupted by a partially submerged vehicle. This again results in the visible deflection of flow. In the main channel, immediately downstream of the bridge, large-scale turbulent structures as a result of secondary circulation are detected with surface velocities progressively increasing to a maximum of 2.14m s$^{-1}$. The uncertainty attached to all calculated velocities

is relatively low with a spatial average across the area of ± 0.15 m s$^{-1}$. Little difference in observed in the uncertainty attached to out-of-bank velocities (± 0.15m s$^{-1}$), and within-channel velocities (± 0.16m s$^{-1}$), illustrating the consistency of the approach.

Figure 4. Images showing a) velocity magnitude and b) standard deviation of measurements calculated by tracking optically visual surface features.

## 4. Discussion

### 4.1 Adoption of feature tracking approach

Application of feature tracking in open channels is dominated by methods operating in the Eulerian frame of reference (e.g. Large-Scale Particle Image Velocimetry). These methods have been widely successful in the characterisation of instantaneous and time-averaged velocities for the determination of flood discharges, with deviations from acoustically derived

measurements of < 10% (Dramais et al., 2011;Jodeau et al., 2008;Muste et al., 2008). Measurements made in the Lagragian frame of reference (e.g. Large-Scale Particle Tracking Velocimetry (LSPTV)), where the path of individual particles are assessed, have been less widely adopted for monitoring high magnitude events. This is despite LSPTV replicating hydraulics accurately with improved performance close to boundaries and in areas experiencing high velocity gradients (Admiraal et al., 2004). Enhanced spatial resolution of measurements may also be possible with lower seeding densities (Detert and Weitbrecht,

2015). Our implementation of the KLT algorithm has demonstrated its potential to generate large volumes of temporally consistent data at a distance of up to 50m. However, feature tracking from non-stationary platforms poses additional challenges



in accounting for errors related to sensor movement and orientation. These challenges, which must be addressed for this approach to be beneficial for monitoring flood flows, are discussed in the following sections.

## 4.2 Transformation errors

Transformation from pixel to world co-ordinates is one of the greatest challenges in generating accurate velocity estimates, even when measurements are conducted in controlled conditions from sensors of known, fixed locations (Lewis and Rhoads, 2015). Specific error associated with rectification can be controlled by ensuring the camera lens is: i) orthogonal to the water surface (e.g. Lewis and Rhoads, 2015); ii) corrected for distortion (e.g. Le Boursicaud et al., 2015); and iii) accurately calibrated using stable GCPs throughout the field-of-view (e.g. Dramais et al., 2011). Unfortunately it is not always possible to maintain the camera lens orthogonal to the water surface whilst capturing flow processes at the scale of interest, which often necessitates oblique image capture. Such oblique image capture may pose methodological difficulties due to far-field objects being poorly resolved relative to those in near-field. Secondly, lens distortion must be removed prior to the implementation of traditional plan-to-plan perspective projection (Le Boursicaud et al., 2015). This can be achieved based on the manufacturer's specifications (e.g. Detert and Weitbrecht, 2015), or through manual calibration (e.g. Tauro et al., 2015a); however residual distortion may persist close to image boundaries. Finally, following internal camera calibration, the success of the transformation depends on the 3-D distribution of GCPs. Distribution of at least four GCPs are required for a two-dimensional transformation (Fujita and Kunita, 2011;Fujita et al., 1998), or minimum six GCPs distributed across the region of interest for a 3D plan-to-plan perspective projection (Jodeau et al., 2008;Muste et al., 2008). For accurate transformation, elevation errors can be minimised by ensuring GCPs are similar to or located parallel to the water surface elevation (Fujita and Kunita, 2011;Jodeau et al., 2008). However, an implicit assumption of this approach is that the planar free surface is horizontal and that free surface undulations are negligible across the frame. Due to the often negligible water surface slopes across the area of interest, errors are typically assumed to be insignificant (Hauet et al., 2008), with previous research indicating that water level errors of ±0.3m result in velocity deviations of < ±5% (Le Boursicaud et al., 2015). A second source of elevation error may be induced by local water level variability as a result of standing waves created by hydraulic jumps, or obstacles. However, previous research (Dramais et al., 2011) has demonstrated that local variability of up to 1m may still have an insignificant impact on stream-wise velocity measurements when images are collected perpendicular to flow. Due to the responsive nature of this Alyth survey to the July rainfall and flood event, the distribution of GCPs was not pre-determined, so despite a total of 15 linear structures within the urban landscape that intersected the water surface being identified as GCPs, spatial coverage is incomplete and availability is temporally variable. While rapid response deployment during floods may therefore introduce errors in the projection that would otherwise be accounted for in planned deployments, the majority of surveys at high discharge will naturally be 'unplanned' and the result of rapid field deployment. Despite this, and the technical challenges of flying surveys during flood periods, the relatively stable transformations achieved throughout the duration of the July 2015 Alyth video presented here demonstrate the utility of the approach.



### 4.3 Accounting for movement

In addition to oblique image capture, camera motion can greatly diminish the precision of any calibration and transformation process. In the case of monitoring fluvial flash floods from UAV platforms, camera motion is inevitable (Tauro et al., 2015a;Tauro et al., 2015b), and this movement should be corrected for on a frame-by-frame basis utilising fixed reference

points (Lewis and Rhoads, 2015). In the approach reported on here, we adopt a methodology to account for these uncertainties and their impacts on subsequent velocity measurements whereby fixed control points are automatically tracked between frames using the KLT algorithm. This is enabled by the distinct image textures of the water surface and the built environment, enabling the precision of the rectification process to be quantified and uncertainty in velocity measurements to be established.

### 5 Conclusions

UAVs have the potential to capture information about the earth's surface in dangerous and previously inaccessible locations. Through image acquisition of flash flood events and subsequent object-based analysis, highly dynamic and oft-immeasurable hydraulic phenomenon may be quantified at previously unattainable spatial and temporal resolutions. The potential for this approach to provide valuable information about the hydraulic conditions present during dynamic, high-energy flash floods has until now not been explored.

In this paper we adopt a novel approach, utilising the KLT algorithm to track features present on the water surface which are related to the free-surface velocity. Following the successful tracking of features, a method analogous to the Vector Correction Method has enabled accurate geometric rectification of velocity vectors. We subsequently explored uncertainties associated with the rectification process induced by unsteady camera movements. The maximum geolocation error is 1.0m, which

provides an indication of the minimum spatial scale over which measurements should be averaged and reported. Significant spatial variability in geo-registration error values are observed with median individual GCP error values ranging from 0.27 – 1.0m. Our analysis eliminates the potential for significant errors being a function of reduced pixel density per unit area as GCP distance increases. Geo-registration errors are relatively stable and occur as a result of persistent residual distortion effects following image correction, especially close to the image boundaries, due to the specified transformation parameters being

sub-optimal. Future approaches should seek to calibrate the internal properties of the camera, rather than adopting manufacturers lens specifications. The apparent ground velocities of the 15 GCPs ranges from 0.05 - 0.13m with no apparent relationship between the distance of the GCP and observed ground velocity. These findings illustrate the relative spatial and temporal stability of the geometric transformation.

The application of this approach to assess the hydraulic conditions present in Alyth Burn during a 1:200 year flash flood (Perth & Kinross Council et al., 2015) resulted in the generation of an average 4.2 measurements $m^{-2}$ at a rate of 508 measurements $s^{-1}$. Analysis of these vectors provided a rare insight into the complexity of channel-overbank interactions during flash floods.



The uncertainty attached to the calculated velocities is relatively low with a spatial average across the area of $\pm$ 0.15 m s$^{-1}$. Within-channel and over-bank uncertainty in velocity estimates is comparable.

Comprehensive and innovative monitoring programmes (e.g. Ip et al., 2006;Quevauviller et al., 2012;Smith et al., 2014) have
5   previously improved understanding of transient, rate limiting processes and catchment dynamics during extreme flash floods (Zanon et al., 2010), Similarly, we anticipate that this methodology will be of great use in quantifying highly transient flood flows within ungauged rivers across a wide range of fluvial environments.

**Acknowledgements**

This work was funded by NERC grant NE/K008781/1 'Susceptibility of catchments to INTense RAinfall and flooding
10   (SINATRA)'. Thanks to Angus Forbes of Angus Forbes Photography www.angusforbes.co.uk for making the UAV footage available.



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



**Tables**

| Optimised Parameter | Frame number | |
| --- | --- | --- |
| | 1 | 140 |
| X (m) | 324566.9 | 324565.8 |
| Y (m) | 748589.7 | 748591.3 |
| Z (m) | 15.2 | 16.4 |
| Yaw (radians) | 0.33 | -0.14 |
| Pitch (radians) | 0.61 | 0.67 |
| Roll (radians) | 0.02 | 0.08 |
| RMSE (px) | 11.4 | 8.3 |

Table 1. Optimised parameters of the distorted camera models

**Figures**

Note: Figure captions are in main body of text





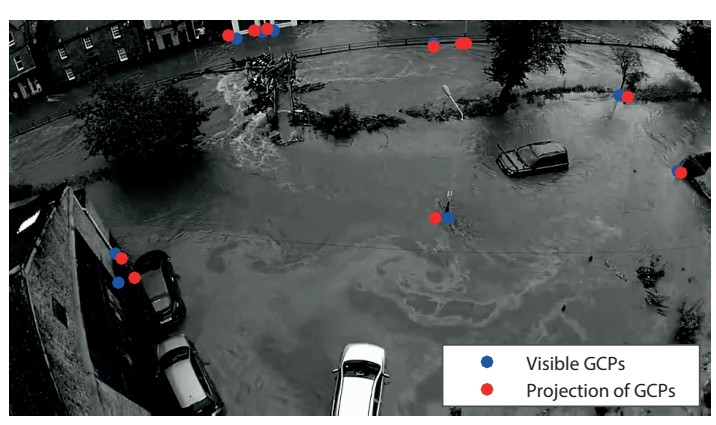





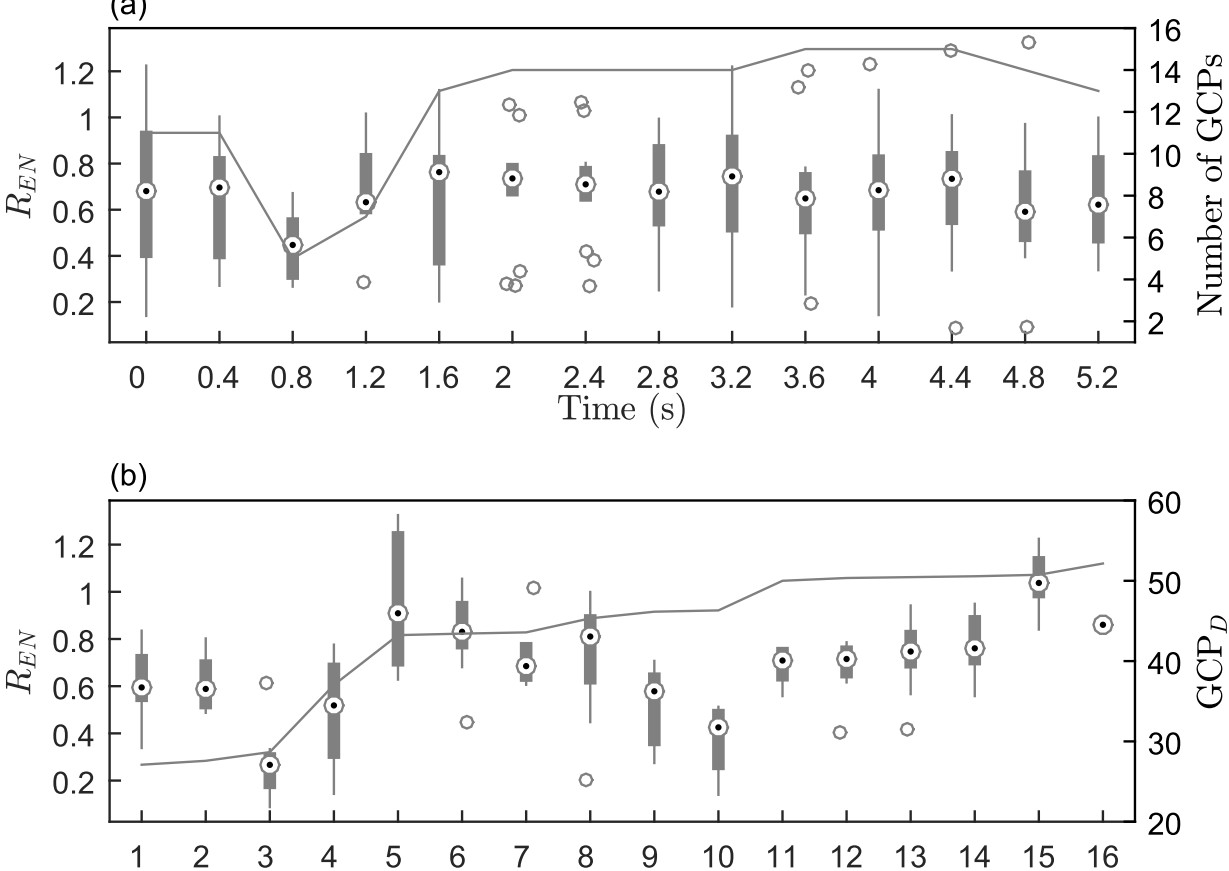

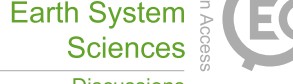

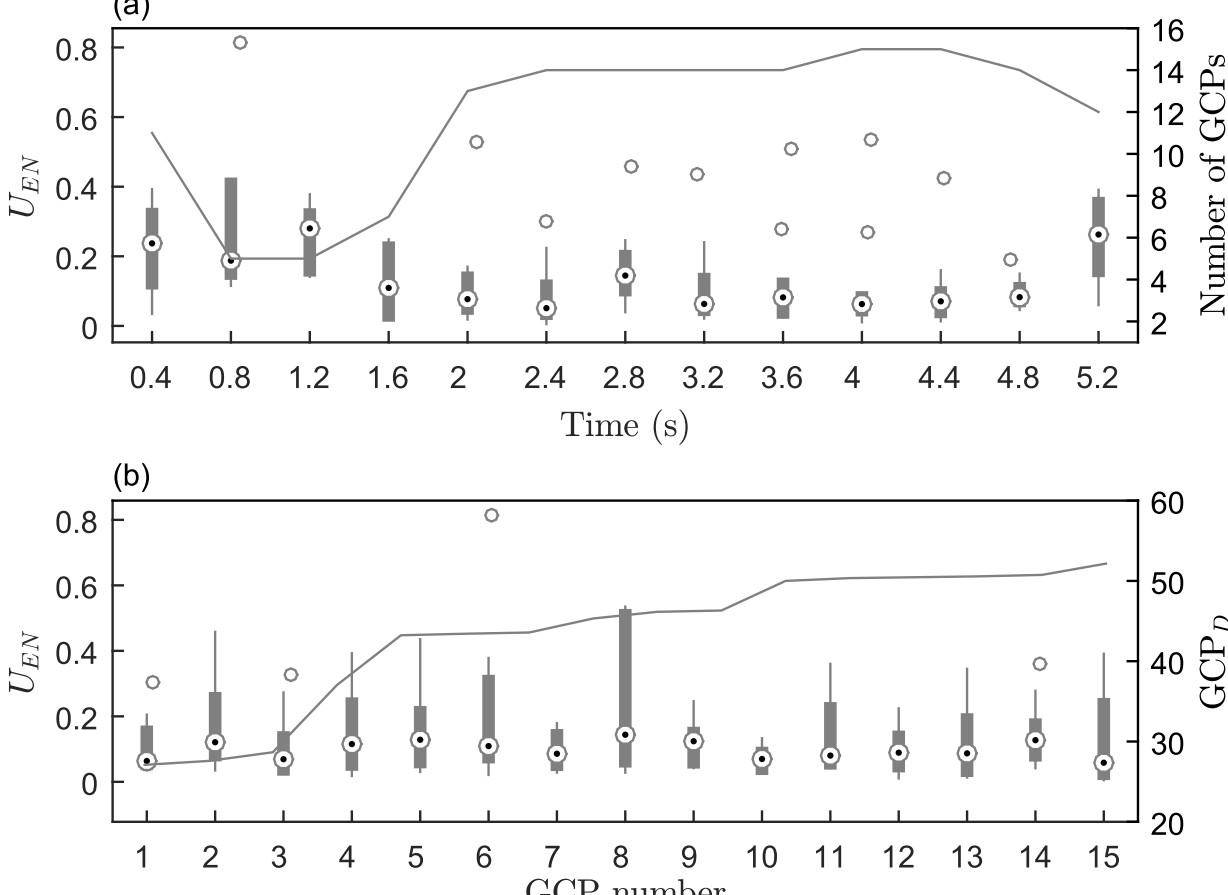





