# Peer review of "Technical Note: Advances in flash flood monitoring using Unmanned Aerial Vehicles (UAVs)"

_Hydrology and Earth System Sciences, 2016_

## Referee Comment (RC1) · Dr. Tauro (Referee) · 8 Mar 2016

In this article, the surface velocity field of the Alyth Burn during a flash flood is reconstructed from images captured by a commercial quadrotor. Velocity estimation is based on a $5.6$ second-footage captured at $25\,\mathrm{Hz}$ at an altitude of $50\,\mathrm{m}$. Images are geo-rectified by a posteriori optimizing the UAV location. Further, virtual velocity vectors are removed by minimizing the difference in the location of actual and projected ground control points. Clearly detectable features in images are tracked using the Kande-Lucas-Tomasi algorithm.

The article is interesting and supports the potential of using UAVs for real time monitoring in complex environments. Remarkably, the Authors demonstrated the feasibility of UAVs for flash flood monitoring through rather low cost equipment, which does not

enable logging of navigation data. While future technological advancements will certainly allow for widespread diffusion of hi-tech platforms, governments and stakeholders could highly benefit from the proposed approach.

The technical note is worth publication in HESS. However, the Authors should consider the following points to improve the manuscript's scope and presentation.

1. The Materials and Methods Section should be simplified to improve on clarity. Paragraph 2.2 is sometimes difficult to read (the coordinates of GCPs and of the UAV starting position can be easily confused). Maybe a flow chart may help in better identifying the image processing steps.

2. Further explanation should be provided on areas presenting poor transformation accuracy (p.5 of Paragraph 2.2). Why were 48% of the original velocity vectors eliminated? How do you plan to improve on that?

3. Details on the video footage should be included. Was it captured while the UAV was remotely piloted or on autonomous navigation? Why didn't you consider flying in the hovering mode to reduce vibrations? How is the camera connected to the platform? Camera gimbals sensibly reduce UAVs vibrations. Image resolution should be included in the Materials and Methods. Also, the footage should be added as supplementary material (I tried to access it on www.angusforbes.co.uk but I was not successful).

4. In my opinion, limitations of the proposed approach lay in the following:

   - Numerous GCPs need to be surveyed in the aftermath of the event
   - Distinct features are necessary to geo-reference the images and apply the tracking algorithm

   The need for GCPs tends to limit the approach to gauged or easily-accessible areas. Conversely, ungauged natural and rather extended regions would be difficult

to monitor. On-site surveying hampers the use of UAVs in wide and impervious areas. How do you plan to compensate for GCPs surveying in such areas?

What is the degree of supervision required by the feature tracking procedure? Do users need to identify the features in images to start the tracking process? I agree with the Authors that Lagrangian-based algorithms may be beneficial in case of low seeding densities. However, they typically require higher supervision by users (a priori information on shape and size of the objects to be tracked). Finally, how long did it take to process the images and extract velocities? How do you plan (if you do) on automating the approach towards real-time analysis?

The article is clearly written and the bibliography is well selected. Since few minor typos can be found, a careful revision of the manuscript is needed.

---

## Referee Comment (RC2) · Anonymous Referee #2 · 10 Mar 2016

This paper describes a technique to measure flow velocities from UAVs films. The approach is based on the Kande-Lucas-Tomasi algorithm to quantify the velocity of patterns on the surface of the flow. Geometric rectification of the velocity vectors is obtained with a method analogous to the vector correction method. Uncertainties are assessed. The proposed method is used to measure surface flow velocities during a 1:200 year flood in Scotland.

Image based technics are now widely used to measure flow velocities. The different methods have proved to be complementary to traditional technics, in particular in dangerous conditions. Besides there are a large number of papers describing such measurements using fixed cameras on the ground or airborne cameras. The development of easy to use and low-cost UAVs offer interesting means to monitor surface flow velocities during high flow conditions and over large areas. Due to the movements of

the camera, there are still some challenges to provide an efficient method to measure hydrodynamic variables.

This paper gives the main points of a method based on UAVs images, laser scan and GNSS system. The approach seems interesting and relevant for a publication in Hydrology and earth system Sciences Discussions. However, there are several points that need to be added in the paper, as well as several modifications are to be made.

First of all, an overview of the method is required, maybe between paragraph 2.1 and 2.2. The different steps of the calculations have to be presented, a small chart could be useful. Furthermore, more details on the algorithms should be given. For example, it is said that "a distorted camera model was generated", could you explain how? I also wonder if the user has to locate the GCPs manually on the pictures. Could you clarify what you call "prominent features" ? The calculation of the flow velocities owing to the first steps of the method should be explained. I was also wondering if the water surface elevation is needed or not.

The method lacks a clear validation step. The obtained flow velocities should be compared with measurements performed with other devices. It could be very interesting to apply the technic on a low flow event to control the results. The proposed validation is only based on optically tracked features, more details are required about this major operation. A small map with the measurement area and the trajectory of the UAV could be helpful in the beginning of the paper.

Could you also specify how you code the different steps (matlab, fortran ?). Are the codes open-source ?

The English is good and the paper is well written. However I have some remarks and questions that must be taken into account to improve the paper:

- In the introduction, you should cite the works dealing with measurements of surface flow velocities from helicopters images. You should also cite the different technics of

image analysis such as LSPIV, LSPTV...

- At the end of paragraph 2.1, the error is for all the directions x, y and z ?

- Some of the Figures (1 and 4 for example) and table 1 are not cited in the text

- The UAV acronym should be make explicit in the abstract (especially for non-english speaking people)

---

## Author Comment (AC2) · 10 May 2016

We wish to thank Anonymous Reviewer #2 for their detailed critique of our paper and for their considered comments. In the following we provide point by point responses to each of the reviewer's comments.

Response to specific comments:

**Comment 1.** *First of all, an overview of the method is required, maybe between paragraph 2.1 and 2.2. The different steps of the calculations have to be presented, a small chart could be useful. Furthermore, more details on the algorithms should be given. For example, it is said that "a distorted camera model was generated", could you explain how? I also wonder if the user has to locate the GCPs manually on the pictures. Could you clarify what you call "prominent features"? The calculation of the flow velocities owing to the first steps of the method should be explained. I was also wondering if the water surface elevation is needed or not.*

**Reply 1.** We agree that a flow chart will be beneficial, in helping to clarify the processing steps and will provide a clear overview of the method. We intend to include one in any revised submission. For brevity, we have not provided details of sub-steps of the approach where other authors have published details of the method. In the example that Reviewer 2 specifies, we adopt the method of Messerli and Grinsted (2015) for the development of the camera model (Page 4 Line 1). Details of this approach and examples of its use are provided within the cited publication. At the start of the processing procedure we ran the KLT algorithm to detect all 'prominent features' (which we define as all features that are extracted using the KLT algorithm). We subsequently manually select the prominent features that we can assign as GCPs. These must be level with the water surface, non-mobile, and clearly visible within the laser scan generated point cloud. These features are then automatically tracked from frame-to-frame using the KLT algorithm. In the final manuscript we will explicitly state how the flow velocities are calculated within Section 2. The method that we adopt here utilises a two-dimensional transformation which assumes that the GCPs are at constant elevation (Page 4 Line 23). By selecting GCPs that intersect with the water surface elevation, we are assuming that the water surface slope is negligible across the image frame. Whilst this assumption is appropriate for relatively small areas such as this application, this may not be appropriate in other applications. We discuss the errors associated with transformation in Section 4.2 and specifically comment on the issue of water surface elevations on Page 8 Lines 19 – 25. This is an issue that warrants further attention and is a current limitation of this method due to the lack of GCPs being established prior to the flood.

**Comment 2.** *The method lacks a clear validation step. The obtained flow velocities should be compared with measurements performed with other devices. It could be very interesting to apply the technic on a low flow event to control the results. The proposed validation is only based on optically tracked features; more details are required about this major operation. A small map with the measurement area and the trajectory of the UAV could be helpful in the beginning of the paper.*

**Reply 2.** Due to the localisation of this flash flood within an ungauged catchment we do not have any data that could be used as validation. Indeed, this was the motivation behind our approach. However, the data presented in Figure 4 clearly replicates observations of how the flow interacts with features and structures which modify the flow path e.g. blocked bridge resulting in flow being diverted along the road, while the reported standard deviations show how stable the velocity field is over the 5.6 seconds of recording. However, we do agree that a

quantitative validation of this approach is required. This is something that we intend to assess in forthcoming research activities. We are happy to provide a map of the UAV trajectory within the revised manuscript.

**Comment 3.** *Could you also specify how you code the different steps (matlab, fortran?). Are the codes open-source?*

**Reply 3.** The entire work-flow is coded in MATLAB R2016a. Although the code is not currently open-source this is something that we seek to achieve in time.

**Comment 4.** *In the introduction, you should cite the works dealing with measurements of surface flow velocities from helicopters images. You should also cite the different technics of image analysis such as LSPIV, LSPTV*

**Reply 4.** We will explicitly mention the different approaches that are available for surface flow measurement, from aerial platforms highlighting different work utilising PIV and PTV methodologies.

**Comment 5**. *At the end of paragraph 2.1, the error is for all the directions x, y and z?*

**Reply 5.** Yes the error that we cite at the end of Section 2.1 relating to the stitching of point clouds and the transformation to real world co-ordinates is the totoal error across the x, y, and z planes. We will make this explicitly clear in the revised text.

**Comment 6.** *Some of the Figures (1 and 4 for example) and table 1 are not cited in the text*

**Reply 6.** This is an oversight on our part and we will rectify this.

**Comment7.** *The UAV acronym should be make explicit in the abstract (especially for non-English speaking people).*

**Reply 7.** We will ensure that the term 'UAV' is properly defined in the abstract

---

## Author Response (AR1)

**Response to Review 1 (Flavia Tauro)**

We wish to thank Dr Flavia Tauro for her insightful comments and wider discussion concerning the application of UAVs and associated methodological developments for the quantification of flood processes. In the following we provide point by point responses to each of the reviewer's comments.

**Comment 1.** The Materials and Methods Section should be simplified to improve on clarity. Paragraph 2.2 is sometimes difficult to read (the coordinates of GCPs and of the UAV starting position can be easily confused). Maybe a flow chart may help in better identifying the image processing steps.

**Reply 1.** We agree that a flow chart would help to clarify the image processing steps. This is now included in this resubmission (Figure 1). We also explicitly specify differences between GCP co-ordinates and camera model co-ordinates to add clarity to Section 2.2 (see marked-up version).

**Comment 2.** Further explanation should be provided on areas presenting poor transformation accuracy (p.5 of Paragraph 2.2). Why were 48% of the original velocity vectors eliminated? How do you plan to improve on that? **Reply 2.** 48% of the vectors were removed due to the apparent movement of the GCPs, or due to unsatisfactory error associated with georectification (Page 6 Lines 15 - 19). This is a result of persistent residual distortion effects following image correction, especially close to the image boundaries, due to the specified transformation parameters being sub-optimal (Page 7 Lines 11 - 13 and Page 11 Lines 4 - 7). This is likely a result of the actual radial distortion parameters of the lens within camera differing slightly from the manufacturers' specification. Assessment of this is not possible without performing manual calibration of the camera, which would have improved the transformational accuracy. To improve the transformational accuracy in future deployments this would be best achieved by using a calibrated camera with minimal lens distortion (Page 11 Lines 6 - 7).

**Comment 3.** Details on the video footage should be included. Was it captured while the UAV was remotely piloted or on autonomous navigation? Why didn't you consider flying in the hovering mode to reduce vibrations? How is the camera connected to the platform? Camera gimbals sensibly reduce UAVs vibrations. Image resolution should be included in the Materials and Methods. Also, the footage should be added as supplementary material (I tried to access it on www.angusforbes.co.uk but I was not successful).

**Reply 3.** The crowd-sourced video footage was collected by a standard DJI Phantom Vision 2 UAV in manual flight mode by a member of the public whose aim was to document the impacts of the floods across the inundated area (Page 3 Line 30 - Page 4 Line 1). The video itself was not collected with the intention of being used for PIV analysis. If this had been piloted by ourselves we would have sought to hover over each region of interest (ROI) for several seconds before moving on. The image resolution of the video footage is 960 x 540 pixels at 25 frames per second (Page 4 Line 1 - 2). We now provide the video footage as Supplementary Material.

**Comment 4.** *In my opinion, limitations of the proposed approach lay in the following:*

- Numerous GCPs need to be surveyed in the aftermath of the event
- Distinct features are necessary to geo-reference the images and apply the tracking algorithm

The need for GCPs tends to limit the approach to gauged or easily-accessible areas. Conversely, ungauged natural and rather extended regions would be difficult to monitor. On-site surveying hampers the use of UAVs in

wide and impervious areas. How do you plan to compensate for GCPs surveying in such areas? What is the degree of supervision required by the feature tracking procedure? Do users need to identify the features in images to start the tracking process? I agree with the Authors that Lagrangian-based algorithms may be beneficial in case of low seeding densities. However, they typically require higher supervision by users (a priori information on shape and size of the objects to be tracked). Finally, how long did it take to process the images and extract velocities? How do you plan (if you do) on automating the approach towards real-time analysis?

**Reply 4.** We agree with the limitations of the approach that Reviewer 1 rightly mentions. This method requires the presence of GCPs that are observable across the camera frame, which must also be accurately surveyed following the event (Page 4 Lines 4 - 7). Within urbanised areas where naturally occurring features are available as GCPs (e.g. lamp-posts, fence lines, walls, etc.) this approach offers a potentially valuable method for quantifying flood flow processes beyond the range of events that can be captured typically using traditional flow measurement techniques. In areas where such GCP features do not exist a different approach would be required, whether it be through the use of lasers, or utilisation of on-board GPS systems in conjunction with additional sensors to facilitate in the transformation process (Page 10 Lines 1 - 2, and Page 10 Lines 10 - 11). Enhancement and adoption of these approaches are key to enable UAVs to be utilised for real-time capture of hydraulic properties of flow in the future (Page 10 Lines 11 - 12). The procedure we adopt requires some supervision. Specifically, during the tracking stage, GCP locations are added, checked and updated every 10 frames as the camera field of view and illumination of the image varies (Page 4 Lines 31 - 32). This procedure ensures that sufficient GCPs are visible throughout the video and that they are still accurately focussed on the correct object in question. In an optimal operation, purpose-built GCPs would be installed across the areas of interest with specific optical characteristics so that (semi-)automatic registration would be possible (Page 10 Lines 7 - 8). The features of interest do not necessarily need to be manually selected prior to tracking. Features across the entire frame are established and it is subsequently possible to specify a ROI, thereby ignoring tracked features beyond this area. Complete automation of the process (no supervision required), camera calibration and tracking of the 5.6 seconds of footage presented here took 87 minutes on a 64-bit Windows OS with a 3.2 GHz CPU and 8 GB installed RAM. The initial development of the master camera model accounted for 29% of this time, with the subsequent tracking, georectification and updates to the camera model accounting for 71% (Page 6 Line 23).

**Response to Anonymous Review #2**

We wish to thank Anonymous Reviewer #2 for their detailed critique of our paper and for their considered comments. In the following we provide point by point responses to each of the reviewer's comments.

**Response to specific comments:**

**Comment 1.** First of all, an overview of the method is required, maybe between paragraph 2.1 and 2.2. The different steps of the calculations have to be presented, a small chart could be useful. Furthermore, more details on the algorithms should be given. For example, it is said that "a distorted camera model was generated", could you explain how? I also wonder if the user has to locate the GCPs manually on the pictures. Could you clarify what you call "prominent features"? The calculation of the flow velocities owing to the first steps of the method should be explained. I was also wondering if the water surface elevation is needed or not.

**Reply 1.** We agree that an overview of the method and flow chart will be beneficial in helping to clarify the processing steps and to provide a clear overview of the method. This is now included in the resubmission (Page 3 Lines 15 - 20 and Figure 1). For brevity, we have not provided details of sub-steps of the approach where other authors have published details of the method. In the example that Reviewer 2 specifies, we adopt the method of Messerli and Grinsted (2015) for the development of the camera model (Page 4 Line 17). Details of this approach and examples of its use are provided within the cited publication. We have now removed the term 'prominent features' from the manuscript and make it clear that all features are utilised are those that are automatically extracted by the KLT algorithm (see marked-up version). GCPs are manually selected from the images (Page 4 Line 30 - Page 5 Line 2). GCPs must be level with the water surface, non-mobile, and clearly visible within the laser scan generated point cloud (Page 5 Lines 2 - 3). These features are then automatically tracked from frame-to-frame using the KLT algorithm. In the resubmitted manuscript we explicitly state how the flow velocities are calculated (Section 2.5). By selecting GCPs that intersect with the water surface elevation, we are assuming that the water surface slope is negligible across the image frame (Page 9 Lines 24 - 26). Whilst this assumption is appropriate for relatively small areas such as this application, this may not be appropriate in other applications. We discuss the errors associated with transformation in Section 4.2.

**Comment 2.** The method lacks a clear validation step. The obtained flow velocities should be compared with measurements performed with other devices. It could be very interesting to apply the technic on a low flow event to control the results. The proposed validation is only based on optically tracked features; more details are required about this major operation. A small map with the measurement area and the trajectory of the UAV could be helpful in the beginning of the paper.

**Reply 2.** Due to the localisation of this flash flood within an ungauged catchment we do not have any data that could be used as validation. Indeed, this was the motivation behind our approach. However, the data presented in Figure 4 clearly replicates observations of how the flow interacts with features and structures which modify the flow path e.g. blocked bridge resulting in flow being diverted along the road, while the reported standard deviations show how stable the velocity field is over the 5.6 seconds of recording. However, we do agree that a quantitative validation of this approach is required moving forward. This is something that we intend to assess in forthcoming research activities. In this resubmission we now provide a map of the UAV trajectory and the camera viewshed within Figure 1.

**Comment 3.** Could you also specify how you code the different steps (matlab, fortran?). Are the codes opensource?

**Reply 3.** The entire work-flow is coded in MATLAB R2016a (Page 4 Line 17). Although the code is not currently open-source this is something that we seek to achieve in time.

**Comment 4.** In the introduction, you should cite the works dealing with measurements of surface flow velocities from helicopters images. You should also cite the different technics of image analysis such as LSPIV, LSPTV

**Reply 4.** We now cite the works of Fujita and Kunita (2011) whom utilise helicopters and LSPIV for flood monitoring (Page 2 Line 17), and we also highlight the different approaches (LSPIV and LSPTV) for image analysis (Page 2 Lines 27 - 29).

**Comment 5**. At the end of paragraph 2.1, the error is for all the directions *x*, *y* and *z*?**

**Reply 5.** Yes the error that we cite at the end of Section 2.1 relating to the stitching of point clouds and the transformation to real world co-ordinates is the total error across the x, y, and z planes. This is clarified in the revised manuscript (Page 4 Lines 10 - 11)

**Comment 6. Some of the Figures (1 and 4 for example) and table 1 are not cited in the text**

**Reply 6.** This is an oversight on our part and this has been rectified in the revised manuscript (see marked up version).

**Comment 7.** *The UAV acronym should be make explicit in the abstract (especially for non-English speaking people).**

**Reply 7.** In the revised manuscript, the term 'UAV' is properly defined in the title, abstract, and first occurrence in the main body of the text (see marked-up version).

[revised manuscript text omitted]

$$\frac{[\Delta X, \Delta Y] = [X_{R+9}, Y_{R+9}] - [X_R, Y_R]}{k = \frac{4}{(F-I)}}$$

$$\frac{[u, v] = [\Delta X, \Delta Y][k]}{(3)}$$

5

10

The degree to which the geo-rectification process is a success is assessed by comparing how the co-ordinates of the surveyed GCPs [N, E] compare to the projected GCP locations  $[N_T, E_T]$ . The residuals [r, s] represent the absolute positional error of the GCPs and provide a direct measure of the accuracy of the geometric transformation from pixel units into geographical co-ordinates, (Figure 1, Section 2.4), given by the Euclidean distance between the actual and projected locations  $R_{EN}$  (Detert and Weitbrecht, 2015):

$$[r,s] = [N_T, E_T] - [N, E]$$
(41)

$$R_{EN} = (r^2 + s^2)^{0.5} \tag{52}$$

15 The degree to which the projection of the GCPs varies over time is assessed by examining the relative changes in the GCP projection locations (m) between the beginning and end of the feature tracking process:

$$[u_{EN}, v_{EN}] = [r_{n+9} - r_n), (s_{n+9} - s_n)]$$

$$U_{EN} = (u_{EN}^2 + v_{EN}^2)^{0.5}$$
(63)
(74)

20

2-D natural neighbour interpolation of the GCP errors is performed, giving spatially distributed estimates of  $R_{EN}$  and  $U_{EN}$ . (Figure 1, Section 2.4).

**2.5 Surface Velocity Calculation**

As with the GCPs, between the nth and n + 9th frame, surface water features are defined and tracked using the KLT algorithm,
with their start and finish positions being stored in pixel units. During this process, features were only tracked if they were within the central 90% of the image. This was necessary to minimise the potential for residual distortion effects to bias measurements, as these were most likely to persist close to the image boundaries (Detert and Weitbrecht, 2015). The start and finish positions (px) of selected surface water features are converted to real-world start and finish co-ordinates i.e. [Xn, Yn] and [Xn+9, Yn+9] respectively. This is again achieved through two-dimensional transformation (Fujita and Kunita, 2011; Fujita et al., 1998), based on the optimised camera models specific to nth and n + 9th frame (Messerli and Grinsted, 2015). This method is analogous to the Vector Correction Method (Fujita and Kunita, 2011) whereby stationary objects yield zero or negligible

velocity values with the movement of surface water velocity vectors being corrected for background image displacement. This

process enables the calculation of 2-D velocities [u, v] following application of a conversion factor k to account for the number of tracked frames I and seconds per frame  $F_{\underline{i}}$

$$\begin{bmatrix} \Delta X, \Delta Y \end{bmatrix} = \begin{bmatrix} X_{n+9}, Y_{n+9} \end{bmatrix} - \begin{bmatrix} X_n, Y_n \end{bmatrix}$$
5
$$k = \frac{1}{(F \ I)}$$
(6)
$$\begin{bmatrix} u, v \end{bmatrix} = \begin{bmatrix} \Delta X, \Delta Y \end{bmatrix} \begin{bmatrix} k \end{bmatrix}$$
(7)

From which the velocity magnitude is obtained:

$$U = \sqrt{u^2 + v^2}$$

10

Velocity [U] measurements in areas defined as having poor transformation accuracy (i.e.  $\geq -\text{Im}R_{EN} \geq 1$  m), or significant apparent movement of the GCPs between frames (i.e.  $U_{EN} \geq 0.3\text{m3 m}$ ) are removed prior to analysis, in addition to tracked features exhibiting minimal displacement (i.e.  $U \leq 0.3\text{m3 m}$ ). This resulted in 48% of the original velocity vectors surface water features being eliminated. Data was not subject to any additional filtering. (Figure 1, Section 2.5).

(8)

**15 **3. Results**

**3.1 Camera Motion**

Using the 20,000 potential solutions, the optimised master camera model was selected based on the minimum square projection error of the GCPs (RMSE). In this instance, the The minimum RMSE of the 20,000 solutions was 11.4px4 px (n = 8). Optimisation of the initial camera model took 25-min (3.2 GHz CPU, 8GB RAM), and accounted for 29% of the total processing time. Following the perturbation of geographical and orientation parameters for each frame, the flight path of the UAV was successfully modelled. (Figure 1, Section 2.3). Cumulative Euclidean distance travelled by the UAV over the 140 frames was 13.2m (2.5m2 m (mean velocity = 2.5 m s-1) whilst the camera rotated on the y-axis by 28o. (Table 1). During the video the RMSE of the optimised camera did not exceed 12.9px9 px with a mean μ of 9.6px6 px and a standard deviation σ of 1.3px3 px.

25

30

Table 1. Optimised parameters of the distorted camera models

**3.2 Positional Accuracy**

[revised manuscript text omitted]

I

I

**Figures**

**Note: Figure captions are in main body of text**

---

## Referee Report (RR1)

[referee-annotated manuscript omitted]